# Leishmaniasis epidemiology in endemic areas of metropolitan France and its overseas territories from 1998 to 2020

Grégoire Pasquier[1], Magalie Demar[2], Patrick Lami[1], Asma Zribi[1], Pierre Marty[3], Pierre Buffet[4], Nicole Desbois-Nogard[5], Jean Pierre Gangneux[6], Stéphane Simon[2], Romain Blaizot[7], Pierre Couppié[7], Louis Thiebaut[1], Francine Pratlong[1], Jean-Pierre Dedet[1], Patrick Bastien[1], Yvon Sterkers[1], Christophe Ravel[1], Laurence Lachaud[1]*, Working Group for the Notification of Human Leishmanioses in France[¶]

1 Université de Montpellier, CNRS, IRD, Centre Hospitalo-Universitaire de Montpellier, MiVEGEC, Laboratoire de Parasitologie-Mycologie, CNR *Leishmania*, Montpellier, France, 2 Laboratoire hospitalo-universitaire de parasitologie-mycologie, Laboratoire associé au CNRL, Centre Hospitalier, Cayenne, France, 3 Laboratoire de parasitologie-mycologie, CHU de Nice, Nice, France, 4 zCentre d'Infectiologie Necker-Pasteur, Institut Pasteur, Paris, France, 5 Laboratoire de parasitologie-mycologie, CHU de la Martinique, Fort de France, France, 6 Département de Parasitologie-mycologie, Université de Rennes, CHU de Rennes, Irset, Inserm, EHESP, Rennes, France, 7 Service de dermatologie, Centre Hospitalier, Cayenne, France

¶ Membership of the working group for the notification of human leishmanioses in France is provided in the acknowledgments.
* laurence.lachaud@umontpellier.fr

**Data Availability Statement:** All relevant data are within the manuscript and its Supporting Information files.

## Abstract

### Background

In France, leishmaniasis is endemic in the Mediterranean region, in French Guiana and to a lesser extent, in the French West Indies. This study wanted to provide an updated picture of leishmaniasis epidemiology in metropolitan France and in its overseas territories.

### Methodology/Principal findings

Leishmaniasis cases were collected by passive notification to the French National Reference Centre for Leishmaniases (NRCL) in Montpellier from 1998 to 2020 and at the associated Centre in Cayenne (French Guiana) from 2003 to 2020. In metropolitan France, 517 autochthonous leishmaniasis cases, mostly visceral forms due to *Leishmania infantum* (79%), and 1725 imported cases (French Guiana excluded), mainly cutaneous leishmaniasis from Maghreb, were recorded. A slight decrease of autochthonous cases was observed during the survey period, from 0.48 cases/100,000 inhabitants per year in 1999 (highest value) to 0.1 cases/100,000 inhabitants per year in 2017 (lowest value). Conversely, imported cases increased over time (from 59.7 in the 2000s to 94.5 in the 2010s). In French Guiana, 4126 cutaneous and mucocutaneous leishmaniasis cases were reported from 2003 to 2020. The mean incidence was 103.3 cases per 100,000 inhabitants/year but varied in function of the year (from 198 in 2004 to 54 in 2006). In Guadeloupe and Martinique (French West Indies), only sporadic cases were reported.

**Funding:** The authors received no specific funding for this work.

**Competing interests:** The authors have declared that no competing interests exist.

## Conclusions/Significance

Because of concerns about disease expansion and outbreaks in other Southern Europe countries, and leishmaniasis monitoring by the NRCL should be continued and associated with a more active surveillance.

### Author summary

Leishmaniasis is a disease due to a protozoan parasite and transmitted by sandfly. In France, this disease is endemic in the Mediterranean region, in French Guiana and to a lesser extent, in the French West Indies. We wanted to provide an updated picture of leishmaniasis epidemiology in metropolitan France and in its overseas territories. In metropolitan France, from 1998 to 2020, 517 autochthonous leishmaniasis cases, mostly visceral forms due to *Leishmania infantum* (79%), and 1725 imported cases (French Guiana excluded), mainly cutaneous leishmaniasis from Maghreb, were recorded. A slight decrease of autochthonous cases was observed during the survey period, from 0.48 cases/100,000 inhabitants per year in 1999 to 0.1 cases/100,000 inhabitants per year in 2017. Conversely, imported cases increased over time (from 59.7 in the 2000s to 94.5 in the 2010s). In French Guiana, 4126 cutaneous and mucocutaneous leishmaniasis cases were reported from 2003 to 2020. The mean incidence was 103.3 cases per 100,000 inhabitants/year but varied in function of the year (from 198 in 2004 to 54 in 2006). In Guadeloupe and Martinique (French West Indies), only sporadic cases were reported. Because of concerns about disease expansion and outbreaks, leishmaniasis monitoring should be continued and associated with a more active surveillance.

## Introduction

Leishmaniasis is a vector-borne disease caused by *Leishmania spp*., an eukaryotic protozoan parasite belonging to the Trypanomastidae family transmitted by infected phlebotomine sandflies. Three main clinical presentations can be observed: cutaneous leishmaniasis (CL), mucocutaneous leishmaniasis (MCL), and visceral leishmaniasis (VL). According to the 2021 World Health Organisation report [1], leishmaniasis remains a major health problem in the Americas, East Africa, North Africa, West Asia, and South-East Asia. Indeed, in 2020, 208,357 CL cases, 12,838 VL cases, and 347 leishmaniasis-related deaths were reported worldwide [1]. However, due to underreporting, it was estimated in 2012 that 0.7 to 1.2 million CL cases, 0.2 to 0.4 million VL cases, and 20,000 to 40,000 leishmaniasis-related deaths occurred per year [2]. Updated data of 2017 [3] indicated a decrease in VL (50,000 to 90,000 estimated cases).

In France, leishmaniasis is endemic in the Mediterranean part of metropolitan France, in French Guiana, and to a lesser extent in the French West Indies. Many imported cases also are recorded. Between 1999 and 2012, the incidence of endemic leishmaniasis in the South of France was 0.21 per 100,000 inhabitants (i.e. a mean number of 22.6 cases per year) [4], and the most common notified clinical form was VL (84.5%). Five endemic foci were described: Pyrénées-Orientales, Cévennes, Provence, Côte d'Azur, and Corsica [5]. Infections seem to occur in two distinct environments: semi-rural hillsides with mixed forests, and the urban/peri-urban region of Marseille [6]. *Leishmania infantum* is the only species involved in the autochthonous leishmaniasis cases in France. The sandfly vectors include *Phlebotomus perniciosus*, the most common, and *Phlebotomus ariasi*, mainly in the Cévennes and Pyrénées-

Orientales foci [7]. These vectors have a seasonal period of activity between June and October [8]. The principal animal reservoir host is the domestic dog (*Canis familiaris*). A review that analysed data on 39,259 tested dogs [9] reported a mean canine seroprevalence of 8%; however, higher seroprevalence (29.6%) was observed in the Cévennes focus [10]. In the same study, PCR screening detected the parasite presence in 79.8% of screened dogs that were either asymptomatic or had canine leishmaniasis. The place of other reservoir hosts, such as red foxes (*Vulpes vulpes*) and cats (*Felis catus*), has been mentioned but remains anecdotal [11].

French Guiana and the French West Indies, two French territories on the American continent, also are endemic areas. In French Guiana, previous studies in 1990 [12] and 2001 [13] reported an incidence of approximately 200 cases/100,000 inhabitants per year, whereas a more recent publication described an incidence of 56 cases/100, 000 inhabitants per year between 2006 and 2013 [14]. Infections follow a seasonal pattern: the highest number of cases is diagnosed in January and the lowest in August [14]. A time series analysis showed that an increase in rainfall is associated with a decrease in the leishmaniasis cases diagnosed two months later [15]. Five species of *Leishmania* are involved: *Leishmania guyanensis*, *Leishmania braziliensis*, *Leishmania amazonensis*, *Leishmania lainsoni* and *Leishmania naiffi*, by decreasing order of prevalence [16]. The digenetic cycle is well known for *L. guyanensis* where the vector is the arboreal phlebotomine *Lutzomyia umbratilis* and the primary reservoir is the two-toed sloth *Choloepus didactylus* [17]. For the other *Leishmania* species, reservoirs are less established: *L. naiffi* has been associated with the nine-banded armadillo (*Dasypus novemcinctus*), *L. lainsoni* with the agouti rodent (*Dasyprocta* spp.), and *L. amazonensis* with the rodent *Proechimys cuvieri* [16]. Only MCL and CL have been described in these overseas regions. *L. braziliensis* and to a lesser extent *L. guyanensis* infections have been associated with MCL [17] that can cause destructive lesions of the nasal and oropharyngeal/laryngeal mucosa [18]. *L. naiffi* infections are often acquired during leisure activities in anthropized coastal areas, whereas infections by the other species are contracted mainly in primary forest regions, during professional activities (soldiers, gold miners) [17]. In the French West Indies (mainly Martinique and Guadeloupe), autochthonous CL and VL cases are sporadically reported [19]. Both *L. infantum* and *Leishmania martiniquensis* are present and both may cause VL [19,20]. The vector is supposed to be *Lutzomyia atroclavata* and the animal reservoir could be black rats (*Rattus rattus*), mongooses (*Herpestes auropunctatus*), marsupials (*Didelphis marsupialis*), and dogs [16].

A surveillance system that covers the South of France, French Guiana and French West Indies was implemented in 1998 with the creation of the French National Reference Centre for Leishmaniases (NRCL) in Montpellier and the associated centre in Cayenne, French Guiana. One of NRCL missions is the epidemiologic monitoring of autochthonous and imported leishmaniasis cases in metropolitan France, French Guiana, and French West Indies. The surveillance system is based on passive and voluntary notification by the medical laboratories or physicians involved in the diagnosis. Each year, these data are analysed and the report is sent to *Santé Publique France* (the French public health agency).

The aim of the present study was to accurately describe leishmaniasis epidemiology in metropolitan France, French Guiana and West French Indies based on the data collected by the NRCL in 22 years (from 1998 to 2020). The data analysis allowed determining the patients' age, sex, place of contamination, and number of lesions and location (for CL).

## Methods

### Ethics statement

Human biological samples and the associated data sent to the French NRCL are registered and declared for research purposes as a biobank for the Centre Hospitalier de Montpellier and the

French National Institute of Health Surveys. No institutional review board approval is required according to the French legislation (article L. 1111–7 du Code de la Santé Publique, article L. 1211–2 du Code de Santé Publique, articles 39 et suivants de la loi 78–17 du 6 janvier 1978 modifiée en 2004 relative à l'informatique, aux fichiers, et aux libertés).

### Data collection

Cases of leishmaniasis were reported to the NRCL by more than 200 sources: university hospitals, general hospitals, health service of the French army, medical laboratories, and general practitioners. The reporting form is available at the following link: https://cnrleish.edu. umontpellier.fr/files/2018/04/Declaration_pub_2011.pdf. The collected data include: age, sex, past medical history especially causes of immunosuppression (HIV, haematological malignancy, solid organ transplant, immunosuppressive therapy, primary immune deficiency), presumed place of contamination, name of the reporting centre, clinical (e.g. number and localization of lesions) and laboratory parameters, method of diagnosis (direct examination, culture, serology, PCR), and species identification if available. Forms are completed anonymously and are analysed in the framework of NRCL public health mission, in cooperation with *Santé Publique France*. Cases were defined as patients with clinical signs and symptoms compatible with at least one positive laboratory test (direct microscopic examination, culture, serology, or real-time PCR). Then, the collected cases were checked and validated by a senior parasitologist before addition to the database. For this study, data collected from 1998 to 2020 (for metropolitan France) and from 2003 to 2020 (for French Guiana and French West Indies) were used.

*Biological diagnosis* was performed according to the protocols in use in each center: direct examination, culture, PCR (and serology for VL). Species identification was based on MLEE or molecular method or MALDI-TOF.

### Statistical analysis

The incidence of autochthonous leishmaniasis was calculated using demographic data from the *Institut national de la statistique et des études économiques*, https://www.insee.fr/; French national institute of statistics and economic studies). The patients' characteristics were expressed as mean (95% Confidence Interval). Multiple pairwise comparisons (pairwise Wilcoxon test) were performed after a significant global test (for quantitative data, ANOVA test if the ANOVA hypotheses were fulfilled, otherwise Kruskal-Wallis test; Fisher's exact test for categorical data). Changes in proportion and incidence over time were assessed by the Cochran-Armitage test for trend. For French Guyana, Chi square test was used to compare relative incidences of leishmaniasis in inland and coastal localities. Statistical analyses were done with the R 3.6.0 software.

### Map design

Maps were designed with the Articque software (https://www.articque.com).

## Results

From 1998 (2003 for French Guyana) to 2020, 6379 leishmaniasis cases were notified to the NRCL (Fig 1): autochthonous leishmaniasis in metropolitan France (n = 517), in French Guiana (n = 4126), and in the French West Indies (n = 11), and also imported leishmaniasis cases from other endemic countries (n = 1725).

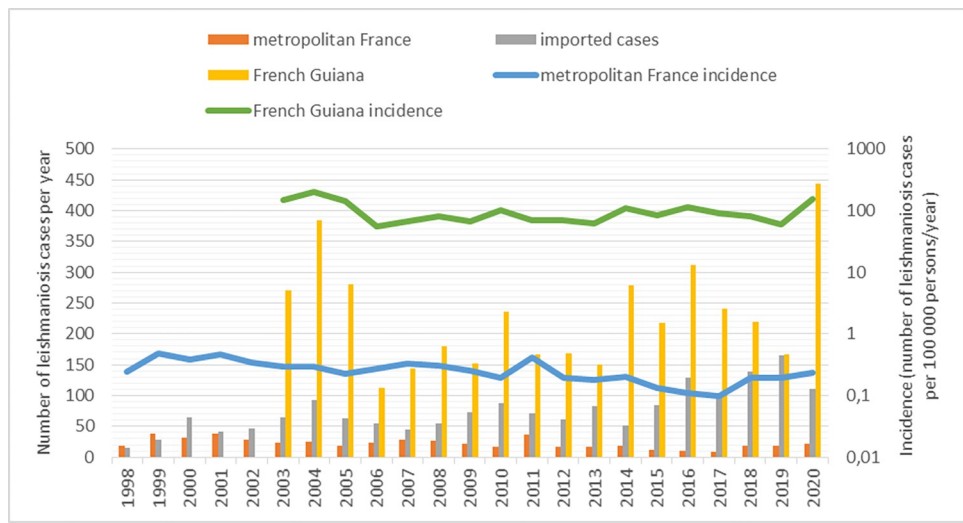

**Fig 1. Number of reported cases to the French National Reference Centre for Leishmaniasis by year.**

Concerning the autochthonous leishmaniasis cases in metropolitan France (n = 517; only caused by *L. infantum*), the men to women ratio was 1.89. The mean incidence in the endemic departments (mainly Alpes-Maritimes, Var, Bouches-du-Rhône, Gard, Hérault and Pyrénées-Orientales) (Fig 2) was 0.26 cases per 100,000 inhabitants/year, and the mean number of cases per year was 22.5 for a population pool of ~9,000,000 inhabitants. The incidence tended to decrease over time from a maximum of 0.48 cases/100,000 inhabitants per year in 1999 to a minimum of 0.1 cases/100,000 inhabitants per year in 2017 (p<0.001) (Fig 1). This reduction should be appraised relative to the decreasing percentage of patients living with HIV (from 53% in 1998 to 9% in 2020; p<0.001) (S1 Fig). The main clinical form was VL (79%, 407/513), followed by CL (18%, 91/513) and MCL (3%, 15/513). Patients with VL were younger than patients with CL and with MCL (35 (32.7–37.3) *versus* 43.7 (38.2–49.3) *versus* 61.8 (55.9–67.7)

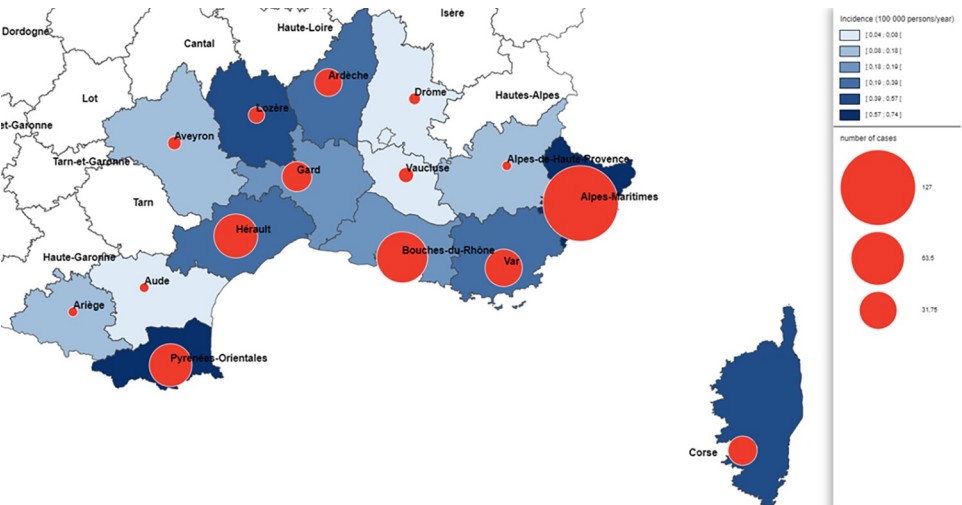

**Fig 2. Distribution of the endemic leishmaniasis cases reported in metropolitan France by *département (i.e.*** French administrative regions) from 1998 to 2020 (n = 380; for 137 others cases département of contamination was missing). Maps were designed with the Articque software (https://www.articque.com).

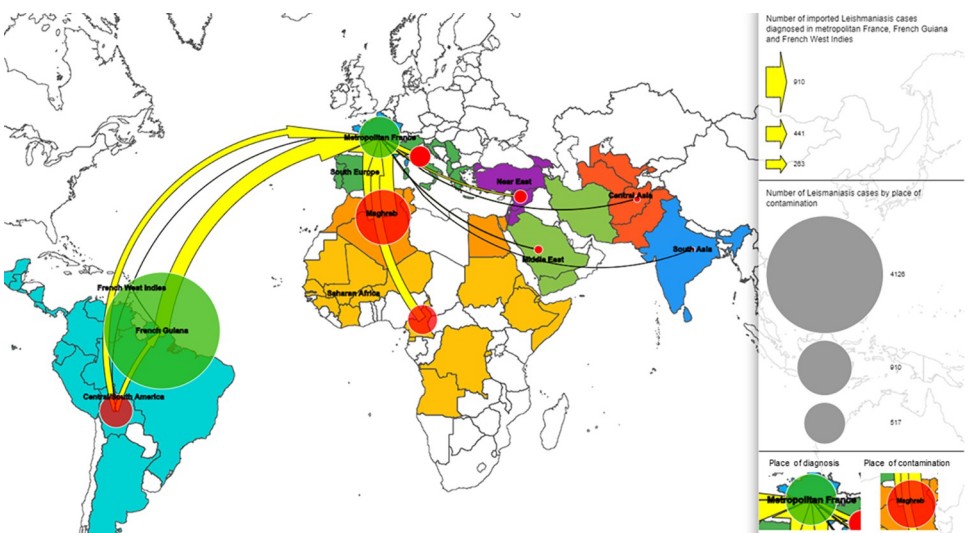

**Fig 3. Origin of the imported leishmaniasis cases in metropolitan France, French Guiana, and French West indies (1998–2020).** Maps were designed with the Articque software (https://www.articque.com).

years, respectively; p<0.001) (S2 Fig). Among the patients with VL, 45% (163/385) were immunosuppressed mainly because of HIV infection (31%, 118/385), immunosuppressive treatment (8%, 29/385), blood malignancies (4%, 16/385), and solid organ transplantation (2%, 9/385). Among the non-immunosuppressed patients with VL, 40% (84/212) were <6-year-old children (S3 Fig).

Most of the diagnosed cases in metropolitan France were imported (n = 1725): 53% (n = 910) from Maghreb, 19% (n = 327) from Central/South America (French Guiana excluded), 15% (n = 265) from sub-Saharan Africa, 8% (n = 133) from Southern Europe, 3% (n = 51) from the Eastern Mediterranean countries (Turkey, Syria, Israel, Jordania) and Georgia, 1% (n = 21) from the Middle East, 1% (n = 13) from Central Asia, and <1% (n = 5) from South Asia (Fig 3). CL was the most frequently encountered clinical form (89%, n = 1542), followed by VL (10% n = 167) and MCL (1% n = 10). The men to women sex ratio was 1.51. The mean number of declared cases per year tended to increase from 59.7 in the 2000s to 94.5 in the 2010s (p>0.001)(Fig 1). Analysis of the species implicated in CL showed that for cases imported from Maghreb, *L. major* was predominant (470/584, 80%) followed by *L. infantum* (62/584, 12%), *Leishmania tropica* (31/584, 5%) and *Leishmania killicki* (21/584, 4%) (Fig 4). *L. major* was predominant also in cases from sub-Saharan Africa (152/166, 92%), whereas *L. tropica* was dominant in case from the Eastern Mediterranean countries (17/27, 55%). Concerning the imported cases from the New World (French Guiana excluded), *L. guyanensis* (136/223, 61%) was the most frequently identified species, followed by *L. braziliensis* (64/223, 29%), and *Leishmania mexicana* (14/223, 6%). For cases from the Old World (metropolitan France included), *L. major* was more associated with multiple lesions (389/580, 67%) than *L. killicki* (5/19, 26%, p = 0.001), *L. tropica* (25/61, 41%, p<0.001) and *L. infantum* (38/142, 27%, p<0.001). Conversely, *L. major* was associated with fewer face lesions (98/494, 20%) than *L. killicki* (10/17, 59%, p = 0.001), *L. tropica* (30/55, 55%, p<0.001) and *L. infantum* (76/128, 59%, p<0.001).

Imported MCL cases (n = 10) were caused by *L. braziliensis* (n = 4 from Central/South America) and *L. infantum* (n = 4 from Southern Europe and n = 2 from Maghreb). The mean age was 48.3 (41.5–55.1) years.

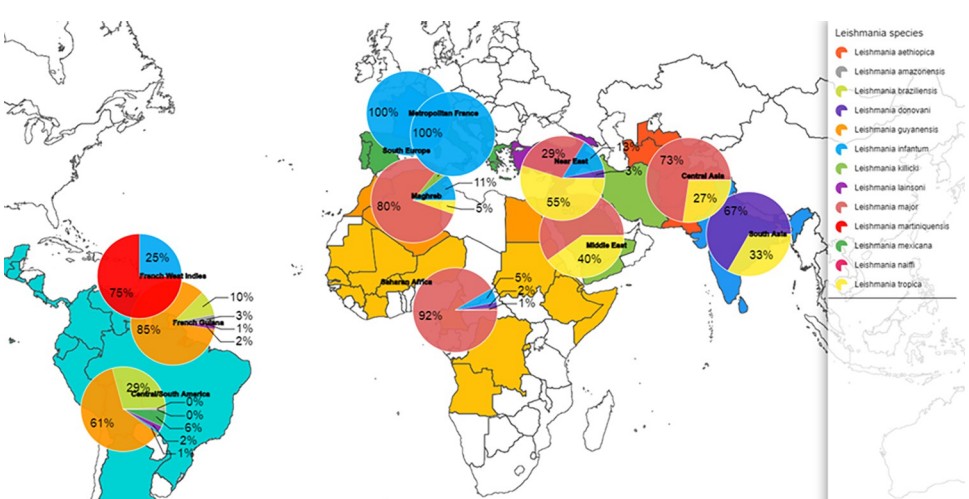

**Fig 4. Distribution of *Leishmania* species according to the place of transmission from 1998 to 2020).** (n = 3393, for 2986 other cases species was missing) Maps were designed with the Articque software (https://www.articque.com).

Imported VL cases (n = 167) were mostly due to infection by *L. infantum* (94%) and by *Leishmania donovani* (6%). Patients mainly contracted the disease in Southern Europe (n = 86, 52%), Maghreb (n = 42, 25%) and sub-Saharan Africa (n = 26, 16%). Their mean age was 38.6 (35.3–41.9) years; 55% of patients were immunocompetent and among them only 26% (22/86) were younger than 6 years of age. In immunosuppressed patients, the causes were HIV (48/71, 68%), immunosuppressive therapy (13/71, 18%), and solid organ transplantation (7/71, 10%).

From 2003 to 2020, 4126 cases of cutaneous and mucocutaneous leishmaniasis were caused by infection in French Guiana (diagnosed in French Guiana or in metropolitan France). The mean incidence was 103.3 cases per 100,000 inhabitants/year and the mean number of declared cases per year was 229 for a mean population of 238,000 inhabitants (Fig 1). The incidence has decreased slightly over time (p<0.001). The men to women ratio was 4.15. No VL case was reported, and MCL cases were exceptional (n = 7). Among the 1969 cases with identified species, *L. guyanensis* was the most frequently detected (85%), followed by *L. braziliensis* (10%), *L. amazonensis* (3%), *L. lainsoni* (2%), and *L. naiffi* (1%). Leishmaniasis cases were less frequent in littoral localities than the in the inland territories and the area along the two main rivers (Oyapoque at the Brazil border and Maroni at the border with Surinam) (Fig 5). Analysis of the data on CL cases from the New World (French Guiana and the rest of Central and South America) showed that patients infected by *L. guyanensis* were more prone to develop multiple lesions (453/913, 50%) compared with those infected by *L. braziliensis* (34/131, 26%, p<0.001) and *L. naiffi* (1/12, 8%, p = 0.046).

In the French West Indies, only few cases were reported. In Guadeloupe, three CL cases and one VL case by *L. infantum* were notified. In Martinique, four CL cases by *L. martiniquensis* and two VL cases, including one caused by *L. martiniquensis*, were recorded. In Saint Martin, one CL case caused by *L. martiniquensis* was recorded.

## Discussion

The creation of the NRCL has allowed the establishment of a surveillance system for indigenous and imported leishmaniasis cases in France. The University Hospital of Cayenne located in French Guiana, a French territory in South America and a leishmaniasis endemic area, has been a NRCL privileged partner for 20 years. This study described and analysed the data

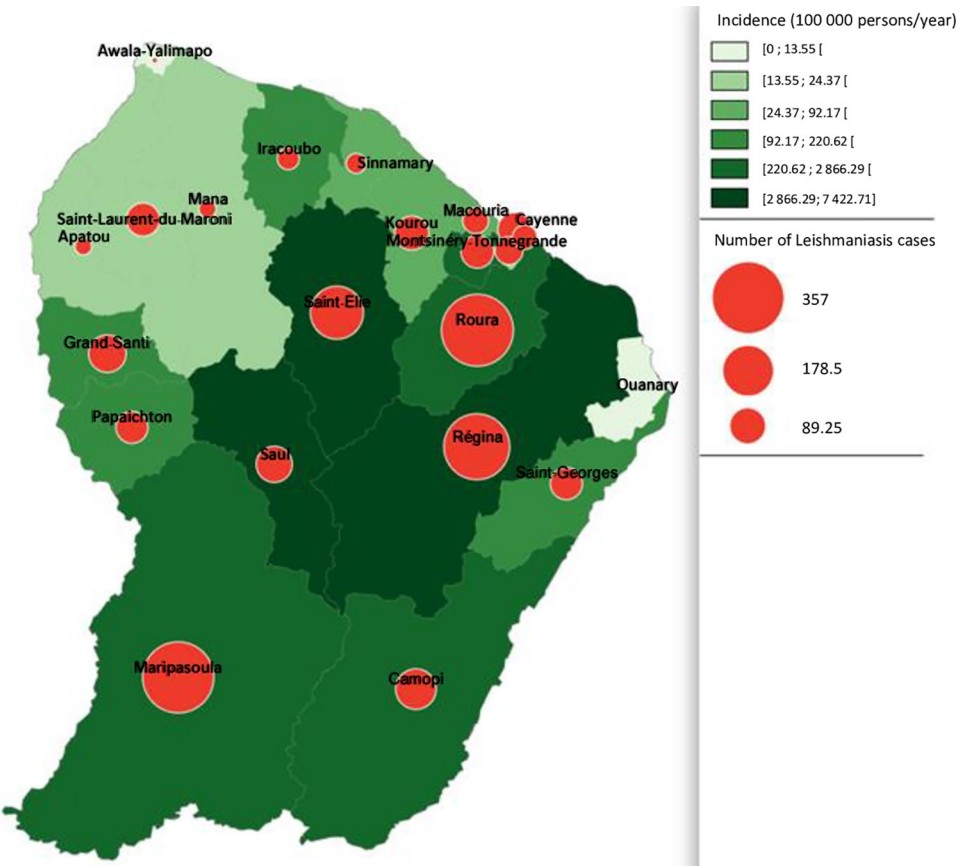

**Fig 5. Distribution of the reported leishmaniasis cases in French Guiana by locality from 2003 to 2020).** (n = 2193, for others 1933 cases, locality was missing) Maps were designed with the Articque software (https://www.articque. com).

collected by the NRCL in 23 years of leishmaniasis surveillance in metropolitan France and overseas territories (n = 6379 cases notified).

In metropolitan France, the incidence of autochthonous cases, mainly VL due to *L. infantum*, decreased from 0.46 in 1999 to 0.1 in 2017 in the hypoendemic area (French Mediterranean coast). This could be linked to the concomitant decrease of patients with HIV and VL (from more than 50% of all leishmaniosis cases in 1998 to less than 10% in 2020) due to the wide implementation of antiretroviral therapy [6]. A similar reduction in leishmaniasis-HIV co-infections was observed in other Southern European countries [21]. On the other hand, no consistent increase of leishmaniasis cases in non-HIV immunosuppressed patients due to new immunosuppressive therapies was observed. The paediatric population (<8 years of age) accounted for half of VL cases in 1986–1987 [22], but only for 24% between 1998 and 2020. Cases in immunocompetent adults represented between 11% and 63% of all reported VL cases, in function of the year. The decline of VL cases may also be partly linked to the widespread use of insecticide-impregnated collars [23] and the anti-leishmaniasis vaccine [24] to control the canine reservoir. The role of human asymptomatic carriers has been suggested, but evidence is lacking [25]. However, some studies suggested that VL foci are distributed more in function of the vector presence than of the canine reservoir density [6]. This rises the concern of the northward spread of phlebotomine vectors caused by global warming [26]. Some authors have warned of possible transmission of leishmaniasis outside Mediterranean coast

such as in the Jura [27] but it remained exceptional. Cases of autochthonous CL or MCL were rare, because *L. infantum* tropism is essentially visceral; however, benign cutaneous forms might be frequently underreported because most of the time they heal spontaneously.

Concerning the imported leishmaniasis cases in metropolitan France (French Guiana excluded), they were predominantly CL caused by *L. major* and *L. tropica* (cases from Maghreb and sub-Saharan Africa) and by *L. guyanensis* and *L. braziliensis* (when imported from South America). The mean number of cases reported per year was 75, with an increase from 15 in 1998 up to 165 in 2019. The first hypothesis is that we have improved the reporting system, the second is that it is possible that there have been changes in migration flows, but we do not have data on this subject. Stays in the Maghreb accounted for 53% (n = 910) of imported cases, another hypothesis would be the increase in vacation trips to this region. In the Old World, 67% of patients with CL due to *L. major* presented multiple lesions, but the face was less concerned compared with other species. This may be of clinical importance because the number and localization of lesions might influence the treatment [3]. These data could be used to revise treatment protocols and to refine the recommendations due to the frequency of multiple lesions [28]. Cases of imported VL due to *L. infantum* were mostly acquired in Southern Europe and Maghreb and accounted only for 10% of all imported cases.

French Guiana, a French territory located in South America, is an endemic zone of cutaneous and mucocutaneous leishmaniasis. The mean incidence was 103.3 cases per 100,000 individuals/year), but it varied in function of the year (from 54 in 2004 to 198 in 2006). This variability may be partly explained by the negative correlation between annual rainfall and number of CL cases [15]. Moreover, most of cases were reported from inland localities, although the population lives mainly along the coastal area (p<0.001). A multicentric prospective study [29] reported that Brazilians represented 59.3% of infected people, mostly gold miners. We can hypothesize that this is one of the reasons that could explain the men to women sex ratio of 4.15 found in French Guiana compared with 1.89 in metropolitan France. The illegal migration of Brazilian gold diggers has been also associated with *L. braziliensis* emergence (from 8.9% in 2006 to 13.0% in 2013) [14]. Indeed, in data from the 1980s, *L. braziliensis* was absent in French Guiana [30]. Moreover, more recently, it was demonstrated that *L. braziliensis* infections are more common in locality with gold mines [14]. However, the NRCL data did not confirm the spread of *L. braziliensis* after 2013. As previously reported [29], patients infected by *L. guyanensis* were more likely to develop multiple lesions compared to those infected by *L. braziliensis*. The precise species identification is crucial for therapeutic adaptation because treatment differs for the two main species in this region: *L. guyanensis* (pentamidine) and *L. braziliensis* (meglumine antimoniate or amphotericin B) [31]. *L. naiffi* was associated with unique lesions, as previously described [17], and with a mild clinical course. The identified risk factors of CL were: living in traditional wooden houses, trips in primary forests, proximity to spring water, and presence of dogs around the house [29]. The other leishmaniasis forms were very uncommon (n = 7 MCL cases in 17 years) or absent (autochthonous VL).

In the French West Indies, leishmaniasis is autochthonous in Martinique, Guadeloupe, and Saint Martin. Sporadic CL and VL cases caused by *L. infantum* and *L. martiniquensis* were reported [19].

Some limitations linked to how data are collected must be acknowledged. The passive surveillance implemented by the NCRL induces underreporting bias compared with active surveillance (17% of underreporting for VL) [6]. Moreover, some data were frequently missing in the reporting forms (lack of species identification in 46% of cases, and absence of clinical information on the number of lesions for 62% of CL cases). Nevertheless, the analysis of national data over a long period of time has updated and shed new light on the epidemiology of

leishmaniasis in France. These results bring moderate concerns about leishmaniasis expansion due to global warming, and highlight the importance of a national surveillance system, such as the NCRL, and of more proactive reporting. Indeed, active notification with direct phone calls to laboratories concerned is more effective but requires more human resources [6]. Future challenges will be to complement these data with epidemiological field studies, for example to monitor canine leishmaniasis and vector distribution in the South of France.

## Supporting information

**S1 Fig. Immune profiles among the patients with visceral leishmaniasis in metropolitan France (n = 385; missing data = 22).**
(TIF)

**S2 Fig. Age of patients according to the leishmaniasis clinical form in metropolitan France (n = 509; 8 missing data); red, mean ± standard deviation.** CL: Cutaneous Leishmaniasis; MCL: Muco-Cutaneous Leishmaniasis; VL: Visceral Leishmaniasis.
(TIF)

**S3 Fig. Age of patients with visceral leishmaniasis according to the immunosuppression type in metropolitan France (n = 403; 4 missing data); red, mean ± standard deviation.** SOT: Solid Organ Transplant; NA: No data Available.
(TIF)

**S1 Table. Number of cutaneous leishmaniasis cases with unique and with multiple lesions according to the causative species.**
(DOCX)

**S2 Table. Number of cutaneous leishmaniasis cases with lesions involving the face according to the species involved.**
(DOCX)

## Acknowledgments

**The Working Group for the Notification of Human Leishmanioses in France** (In alphabetical order and in addition to the main authors) Y Balard (Centre National de Référence des Leishmanioses, Montpellier), P Delaunay and C Pomares (CHU Nice), S Hamane (CHU Saint Louis, Paris), H Yéra (CHU Cochin, Paris), F Foulet (CHU de Créteil), S Houzé (CHU Bichat, Paris), X Iriart (CHU de Toulouse), A Izri (CHU Avicenne, Paris), E Lightburne (Services de Santé des Armées Lavéran, Marseille), Méja Rabodonirina (CHU de Lyon), Coralie L'Ollivier (CHU La Timone, Marseille), Gloria Morizot (Institut Pasteur, Paris), Tiphaine Merguey (CHU de Rennes), Renaud Piarroux (CHU La Timone, Marseille) We warmly thank all the centres that notified leishmaniasis cases to the NRCL over the years, and regret not to be able to name all our correspondents individually.

We thank Pr Nicolas Nagot and Dr Lionel Moulis for their assistance in statistical analysis.

## Author Contributions

**Conceptualization:** Grégoire Pasquier, Laurence Lachaud.

**Data curation:** Grégoire Pasquier, Laurence Lachaud.

**Formal analysis:** Grégoire Pasquier, Laurence Lachaud.

**Investigation:** Magalie Demar, Patrick Lami, Asma Zribi, Pierre Marty, Pierre Buffet, Nicole Desbois-Nogard, Jean Pierre Gangneux, Stéphane Simon, Romain Blaizot, Pierre Couppié, Francine Pratlong, Jean-Pierre Dedet, Patrick Bastien, Yvon Sterkers, Christophe Ravel, Laurence Lachaud.

**Methodology:** Grégoire Pasquier, Laurence Lachaud.

**Resources:** Magalie Demar, Pierre Marty, Pierre Buffet, Nicole Desbois-Nogard, Jean Pierre Gangneux, Stéphane Simon, Romain Blaizot, Pierre Couppié, Louis Thiebaut, Francine Pratlong, Jean-Pierre Dedet, Patrick Bastien, Yvon Sterkers, Christophe Ravel, Laurence Lachaud.

**Supervision:** Laurence Lachaud.

**Writing – original draft:** Grégoire Pasquier, Laurence Lachaud.

**Writing – review & editing:** Grégoire Pasquier, Laurence Lachaud.

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
