## [Decision Letter · Decision Letter 0]

25 Feb 2022

Dear Dr Lachaud,

Thank you very much for submitting your manuscript "Leishmaniasis epidemiology in endemic areas of metropolitan France and its overseas territories from 1998 to 2020" for consideration at PLOS Neglected Tropical Diseases. As with all papers reviewed by the journal, your manuscript was reviewed by members of the editorial board and by several independent reviewers. In light of the reviews (below this email), we would like to invite the resubmission of a significantly-revised version that takes into account the reviewers' comments. 

Major revision is necessary before to be considered for publication. See more detailed reviewers comments and questions below. Please, reply or justify each of the questions/comments.

We cannot make any decision about publication until we have seen the revised manuscript and your response to the reviewers' comments. Your revised manuscript is also likely to be sent to reviewers for further evaluation.

Sincerely,

David Soeiro Barbosa

Associate Editor

Marcelo Ferreira

Deputy Editor

Reviewer's Responses to Questions

**Key Review Criteria Required for Acceptance?**

**Methods**

-Are the objectives of the study clearly articulated with a clear testable hypothesis stated?

-Is the study design appropriate to address the stated objectives?

-Is the population clearly described and appropriate for the hypothesis being tested?

-Is the sample size sufficient to ensure adequate power to address the hypothesis being tested?

-Were correct statistical analysis used to support conclusions?

-Are there concerns about ethical or regulatory requirements being met?

Reviewer #1: The study has a clear objective. However, the methods applied in data analysis do not support most of the discussions and conclusions carried out in the study. Authors must demonstrate their data in relation to the 95% Confidence Interval and not to one standard deviation. A Time Series Analysis must be performed so that statements related to possible trends observed in the study have statistical validity.

Reviewer #2: -Are the objectives of the study clearly articulated with a clear testable hypothesis stated?

Yes

-Is the study design appropriate to address the stated objectives? 

Yes

-Is the population clearly described and appropriate for the hypothesis being tested? 

Yes

-Is the sample size sufficient to ensure adequate power to address the hypothesis being tested? 

Yes

-Were correct statistical analysis used to support conclusions? 

Yes

-Are there concerns about ethical or regulatory requirements being met? 

Yes

Reviewer #3: This article aims to update the epidemiological situation of leishmaniasis in metropolitan France and its overseas territories. The authors uses existing passive surveillance system established for indigenous and imported leishmaniasis cases in the country. The statistical methods used to analyze the available data and address the objectives of the study is appropriate.

**Results**

-Does the analysis presented match the analysis plan?

-Are the results clearly and completely presented?

-Are the figures (Tables, Images) of sufficient quality for clarity?

Reviewer #1: The results are well presented, but they are mostly described data, without entering into some kind of spatio-temporal pattern. Thus, many discussions are not supported by the data presented.

Ex1: This may be due to new migratory dynamics and/or to the improvement of the notification system. Sojourns in

the Maghreb region were at the origin of 53% (n=910) of imported cases (53%, n=910).

Ex2: A multicentric prospective study (28) reported that Brazilians represented 59.3% of infected people, mostly gold

miners. This could explain the men to women sex ratio of 4.15 found in French Guiana compared with 1.89 in metropolitan France.

The study states that a reduction in incidence from 0.48 to 0.1 is a slight reduction. Depending on the graphic scale, it may seem small, but in proportional terms it is a reduction of 4.8 times. Is it statistically different? The time series decomposition could state whether the trend is decreasing or stability.

Reviewer #2: - Does the analysis presented match the analysis plan?

Yes

-Are the results clearly and completely presented?

The reviewer suggests an implementation on the presentation of results

-Are the figures (Tables, Images) of sufficient quality for clarity?

Yes

Reviewer #3: The results and analyses are clearly presented.

**Conclusions**

-Are the conclusions supported by the data presented?

-Are the limitations of analysis clearly described?

-Do the authors discuss how these data can be helpful to advance our understanding of the topic under study?

-Is public health relevance addressed?

Reviewer #1: Many of the conclusions and discussions of the study are made from data that were not generated by the study or are not supported by the data presented. For example: This is related to the return from holydays in this region.

The authors make the following statement:Moreover, cases showed a spatial heterogeneity. 

What result or analysis can support this discussion? Only the visualization of maps would not allow this statement. Clustering analyses, such as Moran's Index, Moran's Local Index or spatial scanning should be used, if any discussion related to heterogeneity wants to be carried out.

Second paragraph of the discussion, at the beginning only reaffirms results, without discussing.

Page 19: Concerning the clinical features, L. guyanensis was more often associated with multiple cutaneous lesions than L.

braziliensis, as mentioned elsewhere (28). Review the sentence. In some parts of the text the percentage value - such as 10%, is displayed twice.

The authors make the following statement: Nevertheless, the analysis of data over a long period of time allowed a reliable

assessment of the disease spatiotemporal changes. A descriptive analysis of the data does not support this result. The temporal analysis was discussed in a descriptive way. And the spatial analysis was performed only with thematic maps. A spatiotemporal pattern analysis should be conducted.

Reviewer #2: -Are the conclusions supported by the data presented?

Yes

-Are the limitations of analysis clearly described?

Yes

-Do the authors discuss how these data can be helpful to advance our understanding of the topic under study?

Yes

-Is public health relevance addressed?

Yes

Reviewer #3: The conclusion is supported by the data and analysis. The limitation of the data and analysis is clearly described. The paper also pinpoints the potential use of the presented information for public health measures.

**Editorial and Data Presentation Modifications?**

Reviewer #1: There is a need for changes in methodology and analysis, so that the results are more clearer and the study discussions are related to its findings. Therefore, my opinion is for the rejection of the paper, requesting that changes be made in the structure of the document, for a future new submission in the journal.

Reviewer #2: (No Response)

Reviewer #3: This paper presents important epidemiological updates about the status of leishmaniasis (Cutaneous; Mucocutaneous and visceral forms), parasite species, demographic characteristics and their distributions despite the limitedness of data availability and use of a passive surveillance data. The authors may consider the following minor edits:

The term Tegumentary leishmaniais is less commonly used and not very well understood especially in the old world. Therefore, please consider replacing by cutaneous and mucocutaneous leishmaniasis where ever it's used in the manuscript.

The leishmaniasis situation in traditionally non-endemic areas of metropolitan France has not been highlighted or discussed; however, as the paper attempts to discuss expansion of leishmaniasis, its important to include some updates in this regard, e.g., reports of leishmaniasis in the Jura region..

Place of contamination ...in this paper is used as place of origin of infection or disease. Whereas in English as this could be confusing and as its commonly used, it's preferred to use ..Place where infection or disease acquired, Place where infection/disease contracted or place of transmission;

Please replace Near East by middle east or the exact countries referred as this term is not standard and no more used and may not be understood by many readers of the article/journal. For the titles of the figures, e.g., Fig 4, 5 .. consider replacing 'repartition' by 'distribution' ....

On page 18, last sentences of the second paragraph, please consider rephrasing as follows : These data could be used to revise treatment protocols and .....

**Summary and General Comments**

Reviewer #1: The greatest strength of the study is a large database, with very important information in the understanding of leishmaniasis. The weakness of the study is that it does not apply a methodology that supports the discussions presented.

Reviewer #2: The manuscript on the "Leishmaniasis epidemiology in endemic area of metropolitan France and its overseas territories from 1998 to 2020" is very interesting and well structured. However, the presentation of data and its analysis sometimes appears incorrect because the missing /incomplete data are not reported. For the section Results missing data are reported only in the Supporting Information. For this the reviewer proposes to improve methods and results.

The manuscript is for “Minor Revision”.

Specific comments 

Pag. 4, Line15: Correct “The animal reservoir host…” in “The principal animal reservoir host…”. 

Pag. 6, Lines 8-16: The AA should briefly summarise the methods used for human diagnosis and Leishmania species identification. The link for the reporting form is important but it is difficult to the readers look for the specific diagnostic details and methods applied.

Pag. 6, Lines 16-19: As suggested by WHO, diagnosis of VL and CL are different. VL diagnosis is a combination of a parasitological (microscopic, culture or molecular) and a serological technique. In addition, the AA don’t give any information about the diagnostic gene/s primers for real-time PCR.

Results: In the whole section missing data are not reported in the manuscript text except in the Supporting Information. The AA should improve the manuscript text furnishing both missing and incomplete data in the results. Similarly, the total of Leishmania species identification samples (strains and bioptic samples) should be indicated.

Pag. 9, Lines 4 and 13: the geographical term of Near East is often overlapping with Middle East. Middle East is the geographical area better identified therefore the AA should clarify (as reported in the map) regions remaining in the Near East.

Refs 16 and 17. The AA should check these 2 references because they appear reverse as citation in the text (Pags 9 and 10).

Reviewer #3: This article aims to update the epidemiological situation of leishmaniasis in metropolitan France and its overseas territories. Considering the limitation of the passive surveillance, the paper has analysed and presented the trend in the three forms of the disease over the years, the distribution of cases and parasite species over the geographic areas focusing mainly on the endemic areas and available demographic variables. As the paper attempts also to discuss the potential expansion of the disease with changing environmental and climatic changes, its important to include some highlights/reports from non-endemic areas and also to elaborate on how or what type of active surveillance can be implemented as this is one of their final conclusion and recommendation.
---

## [Decision Letter · Decision Letter 1]

16 Aug 2022

Dear Dr Lachaud,

We are pleased to inform you that your manuscript 'Leishmaniasis epidemiology in endemic areas of metropolitan France and its overseas territories from 1998 to 2020' has been provisionally accepted for publication in PLOS Neglected Tropical Diseases.

Best regards,

David Soeiro Barbosa

Academic Editor

Marcelo Ferreira

Section Editor

Please, include changes or justify considering the new suggestions.

Reviewer's Responses to Questions

**Key Review Criteria Required for Acceptance?**

**Methods**

-Are the objectives of the study clearly articulated with a clear testable hypothesis stated?

-Is the study design appropriate to address the stated objectives?

-Is the population clearly described and appropriate for the hypothesis being tested?

-Is the sample size sufficient to ensure adequate power to address the hypothesis being tested?

-Were correct statistical analysis used to support conclusions?

-Are there concerns about ethical or regulatory requirements being met?

Reviewer #1: -Are the objectives of the study clearly articulated with a clear testable hypothesis stated? YES

-Is the study design appropriate to address the stated objectives? YES

-Is the population clearly described and appropriate for the hypothesis being tested? YES

-Is the sample size sufficient to ensure adequate power to address the hypothesis being tested? YES

-Were correct statistical analysis used to support conclusions? YES

-Are there concerns about ethical or regulatory requirements being met? YES

Reviewer #3: (No Response)

**Results**

-Does the analysis presented match the analysis plan?

-Are the results clearly and completely presented?

-Are the figures (Tables, Images) of sufficient quality for clarity?

Reviewer #1: -Does the analysis presented match the analysis plan? YES

-Are the results clearly and completely presented?

I suggest changing the term slight in lines 221 and 318.

In line 359 the authors used the term "it hugely varied", in percentages that are very similar.

-Are the figures (Tables, Images) of sufficient quality for clarity? YES

Reviewer #3: (No Response)

**Conclusions**

-Are the conclusions supported by the data presented?

-Are the limitations of analysis clearly described?

-Do the authors discuss how these data can be helpful to advance our understanding of the topic under study?

-Is public health relevance addressed?

Reviewer #1: All requested changes have been made and the conclusion is adequate.

Reviewer #3: (No Response)

**Editorial and Data Presentation Modifications?**

Reviewer #1: The requested changes are very specific. I entered MINOR REVISION, but it could be ACCEPTED.

I believe that in percentage terms the term slight is not adequate. Readers of the article may start to replicate the term SLIGHT, without understanding the percentage context in future citations of the paper, which in my view is an inadequate understanding of the epidemiological change from 0.46 to 0.1.

Reviewer #3: (No Response)

---

## [Editor Report · Acceptance letter]

23 Sep 2022

Dear Pr Lachaud,

We are delighted to inform you that your manuscript, "Leishmaniasis epidemiology in endemic areas of metropolitan France and its overseas territories from 1998 to 2020," has been formally accepted for publication in PLOS Neglected Tropical Diseases.

Best regards,

Shaden Kamhawi

co-Editor-in-Chief

Paul Brindley

co-Editor-in-Chief
